# Regulation of mRNA translation during mitosis

**Marvin E Tanenbaum[1], Noam Stern-Ginossar[1,2,3], Jonathan S Weissman[1,2,3], Ronald D Vale[1]\***

[1]Department of Cellular and Molecular Pharmacology, Howard Hughes Medical Institute, University of California, San Francisco, San Francisco, United States; [2]Center for RNA Systems Biology, Berkeley, United States; [3]California Institute for Quantitative Biomedical Research, University of California, San Francisco, San Francisco, United States

**Abstract** Passage through mitosis is driven by precisely-timed changes in transcriptional regulation and protein degradation. However, the importance of translational regulation during mitosis remains poorly understood. Here, using ribosome profiling, we find both a global translational repression and identified ∼200 mRNAs that undergo specific translational regulation at mitotic entry. In contrast, few changes in mRNA abundance are observed, indicating that regulation of translation is the primary mechanism of modulating protein expression during mitosis. Interestingly, 91% of the mRNAs that undergo gene-specific regulation in mitosis are translationally repressed, rather than activated. One of the most pronounced translationally-repressed genes is Emi1, an inhibitor of the anaphase promoting complex (APC) which is degraded during mitosis. We show that full APC activation requires translational repression of Emi1 in addition to its degradation. These results identify gene-specific translational repression as a means of controlling the mitotic proteome, which may complement post-translational mechanisms for inactivating protein function.

*For correspondence: vale@cmp.ucsf.edu

**Competing interests:** The authors declare that no competing interests exist.

## Introduction

Genome-wide microarray, RNA sequencing (RNA-seq), and protein-based mass spectrometry studies have revealed changes in the abundance of hundreds of proteins during the cell cycle (*Cho et al., 2001*; *Whitfield et al., 2002*; *Aviner et al., 2013*; *Grant et al., 2013*; *Lane et al., 2013*; *Stumpf et al., 2013*; *Ly et al., 2014*), many of which are specifically expressed in G2 phase and mitosis (G2/M). Transcriptional regulation plays an important role in this temporal expression pattern, as many genes show cell cycle-stage specific expression of their mRNA level (*Cho et al., 2001*; *Whitfield et al., 2002*). Regulated protein degradation also plays a key role in sculpting the proteome during the cell cycle, particularly at the end of mitosis when a large set of proteins is ubiquitinated by the E3 ubiquitin ligase, the Anaphase Promoting Complex (APC), and then degraded by the proteasome (*Peters, 2006*).

In addition to regulation of transcription and protein degradation, changes in translation efficiency (TE) also can modify protein abundance during the cell cycle. Translational regulation plays a particularly important role during the specialized cell division cycles of meiosis and early embryonic development, since transcription is largely silent at this stage (*Mendez and Richter, 2001*; *Groisman et al., 2002*; *Tadros and Lipshitz, 2009*; *Weill et al., 2012*). For example, initial work on meiosis in vertebrate oocytes revealed that lengthening of the poly(A) tails of dormant mRNAs through polyadenylation results in translational activation, which plays a critical role in meiotic progression (*Weill et al., 2012*; *Subtelny et al., 2014*). While somatic cells appear to use transcriptional regulation as a major mechanism for modulating protein expression during the cell cycle, translational regulation

**eLife digest** The human body contains billions of cells, most of which formed via a process called mitosis in which a single cell divides to produce two new daughter cells. Actively dividing cells pass through a series of events (or phases) that are collectively known as the cell cycle. These phases allow the cell to grow in size, copy its genetic material, and then make preparations for cell division before taking the final decision to divide.

Many proteins are involved in regulating the cell cycle and each protein has a particular role in specific phases. The levels of these proteins in cells may change during the cycle, which is often crucial to allow the cell to progress to the next phase. For example, cells need a group of proteins called the anaphase-promoting complex (or APC for short) to destroy other specific proteins at the end of mitosis.

Another way in which the amount of protein in a cell can be adjusted is by controlling how much new protein is made during a process known as translation. During this process, a molecule called a messenger RNA (mRNA)—which contains information copied from a particular gene—is used as a template to assemble a new protein. However, it is not clear whether regulation of translation is involved in control of the cell division.

Tanenbaum et al. now address this question using a technique called ribosome profiling to measure the translation of individual mRNA molecules. The experiments analysed the changes in protein production before, during and after mitosis. The overall level of translation of all the mRNAs was about 35% lower during mitosis. However, some mRNAs in particular experienced a very large reduction in the level of translation (between three- and ten-fold less than the levels before mitosis).

One example of an mRNA whose translation is turned off in mitosis is the mRNA that makes a protein called Emi1. It is known from previous work that Emi1 inhibits the activity of the APC. Therefore, Emi1 needs to be inactivated in mitosis so that the APC can become active and promote progression to the next phase of the cell cycle. It was previously shown that Emi1 is destroyed during mitosis to allow the APC to operate. Tanenbaum et al. found that translation of the Emi1 mRNA must also be suppressed during mitosis in order to keep Emi1 protein levels very low and allow the APC to become fully active. These findings uncover a new role for the control of protein production in regulating the cell cycle. The next challenge will be to find out whether suppression of translation is also used in other biological systems where proteins need to be rapidly inactivated.

has been described in somatic cells as well (*Sivan and Elroy-Stein, 2008*; *Novoa et al., 2010*; *Kronja and Orr-Weaver, 2011*; *Aviner et al., 2013*; *Stumpf et al., 2013*). As in meiosis, changes in the poly (A) tail length can occur during G2 phase of the mitotic cell cycle (*Novoa et al., 2010*), but these effects appear to be less important for translational regulation during the somatic cell cycle than during early embryonic development (*Subtelny et al., 2014*). Perhaps the most striking example of translation regulation in the somatic cell cycle occurs during mitosis, when translation is thought to be globally repressed (*Fan and Penman, 1970*). This global translational repression appears to be sequence-independent, and likely affects most mRNAs (*Fan and Penman, 1970*). While such global translational repression was observed many decades ago, its functional significance remains unknown. Several studies suggested that this global translational repression may facilitate the selective synthesis of a small number of proteins that can escape global inhibition through a non-canonical, mRNA cap-independent translation initiation mechanism dependent on internal ribosome entry sites (IRESes) (*Cornelis et al., 2000*; *Pyronnet et al., 2000*; *Qin and Sarnow, 2004*; *Schepens et al., 2007*; *Wilker et al., 2007*; *Marash et al., 2008*; *Ramirez-Valle et al., 2010*). However, recent work challenged the view that translation in mitosis is mediated to a significant extent by IRESes, and instead found that canonical, cap-dependent translation dominates in mitosis (*Shuda et al., 2015*). Therefore, it is unclear whether IRES-dependent translation represents a general mechanism of translational regulation during mitosis, and whether such IRES-dependent translational activation represents, a minor, or the dominant mechanism of gene-specific translational regulation during mitosis.

The recent development of ribosomal profiling, a method that uses deep sequencing of ribosome-protected mRNA fragments to quantify ribosome occupancy on individual mRNAs, allows the TE of single mRNA species to be examined at a system-wide level (*Ingolia et al., 2009*, *2011*). A recent

study applied this technology to investigate the mechanism of cell cycle-dependent translational control and found hundreds of genes that show changes in TE during the cell cycle (*Stumpf et al., 2013*). Many mRNAs showed altered translation rates in mitosis when compared to either G1 or S phase cells. However, this study did not include analysis of G2 phase cells, which is required to determine whether observed changes are mitosis-specific, as G2 phase cells are very similar to mitotic cells, but very different from G1- or S-phase cells, at least with respect to mRNA levels (*Cho et al., 2001*; *Whitfield et al., 2002*). Thus, it is imperative to compare mitotic cells with both G2 cells (pre-mitotic entry) and G1 cells (post-mitotic exit) in order to identify the translational changes that occur specifically during mitosis. It is also critical to obtain a minimally perturbed population of mitotic cells. Previous studies of mitotic translational regulation relied on synchronizing cells in mitosis with microtubule targeting drugs (*Fan and Penman, 1970*; *Pyronnet et al., 2001*; *Stumpf et al., 2013*), but recent work has shown that translation rates are dramatically affected by treatment with such drugs (*Sivan et al., 2011*; *Coldwell et al., 2013*), thereby complicating the annotation of mitosis-specific translation effects.

Here, using metabolic labeling, combined with ribosome profiling and a tight cell cycle synchronization protocol that does not require microtubule drug treatment, we have identified two distinct translational programs that occur during mitosis. First, using metabolic labeling of non-transformed RPE-1 cells, we find a modest (~35%) global translational repression of the bulk of mRNAs during mitosis, which is consistent with findings in other studies (*Fan and Penman, 1970*; *Bonneau and Sonenberg, 1987*; *Pyronnet et al., 2001*). In addition to this modest global repression, using ribosomal profiling, we identify a subset ~200 of mRNAs that show much larger (>threefold), gene-specific changes in their TE during mitosis. The large majority of the latter group of mRNAs are translationally repressed at mitotic entry and then translationally re-activated at mitotic exit, highlighting the precise temporal specificity of this gene-specific translational regulation. Thus, translational repression, rather than IRES-dependent activation of mRNA translation, is the dominant method of gene-specific translational regulation in mitosis, although minor effects of IRES-dependent translation cannot be ruled out. Follow-up studies on one of these translationally repressed genes, Early mitotic inhibitor 1 (Emi1), a potent inhibitor of the APC, reveals that translational repression in mitosis is important to prevent new Emi1 protein synthesis at a time when the existing Emi1 protein pool is degraded. These results lead to a model in which the combined activities of protein degradation and translational repression ensures the complete removal of Emi1 protein, which enhances APC activation and the degradation of APC substrates at the end of mitosis. These results provide the first genome-wide view of the translational changes that occur specifically in mitosis and reveal that translational regulation can enhance the efficiency of post-translational protein inhibition, which may represent a more general function for translational repression.

## Results

### Comparison of translation regulation and regulation of mRNA levels during mitosis

To study mRNA translation at a genome-wide level during mitosis, we used ribosome profiling, a recently developed technique that enables precise measurements of translation of each mRNA in the cell (*Ingolia et al., 2009*, *2011*). In ribosome profiling, the small fragments of mRNA (~30 nt) that are associated with a ribosome (called the ribosome footprints [FPs]) are isolated and quantitatively analyzed by deep sequencing. This sequence information allows the calculation of the average number of ribosomes per mRNA, which reports on the TE of each mRNA. In parallel, we analyzed total mRNA content by RNA-seq. As many cell cycle-regulated pathways are deregulated in cancer, we used a non-transformed human epithelial cell line, RPE-1, for these studies, as these cells can be precisely synchronized in the cell cycle (see below).

We synchronized RPE-1 cells in late G2 (G2), mitosis (M) or early G1 (G1) with a specific small molecule CDK1 inhibitor, RO-3306, using a previously established synchronization protocol (*Vassilev et al., 2006*) (*Figure 1A*). In this protocol, cells are first arrested in G2 using the CDK1 inhibitor. CDK1 is largely inactive during G2 and only becomes activated at the end of G2/prophase (*Jackman et al., 2003*; *Gavet and Pines, 2010*). Thus, the inhibitor prevents the progression out of G2, but the low CDK1 state in the presence of the CDK1 inhibitor reflects the normal G2 phase in unsynchronized cells (*Jackman et al., 2003*; *Gavet and Pines, 2010*). To release cells from G2, the CDK1 inhibitor is

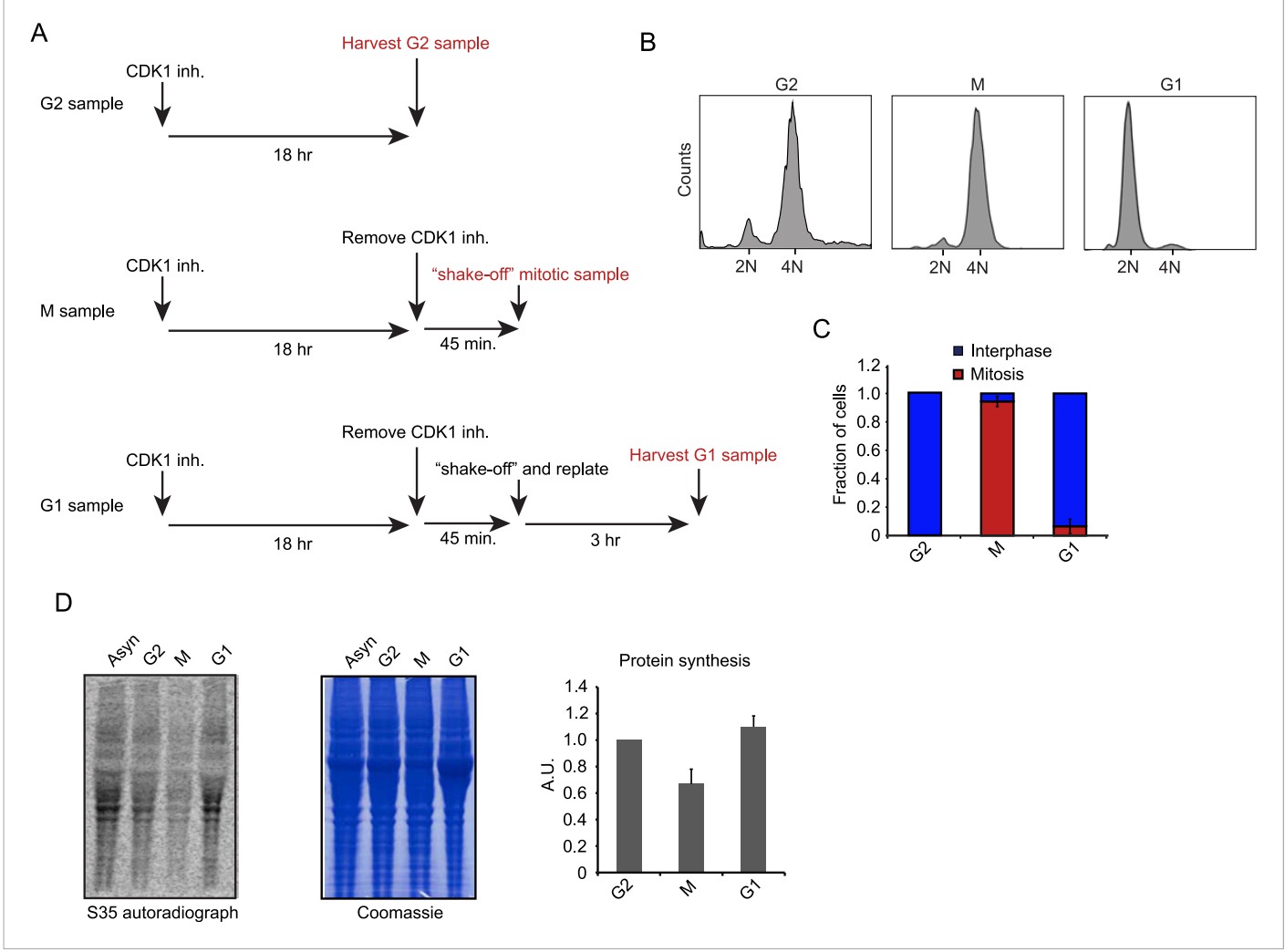

**Figure 1**. Cell synchronization and analysis of translation efficiency during the cell cycle. (**A**) Schematic overview of RPE-1 cell synchronization protocol. RO-3306 (6 μM) was used as the CDK1 inhibitor. (**B**, **C**) G2, M and G1 samples were prepared as outlined in (**A**). (**B**) FACS analysis (Hoechst staining of DNA) reveals that the samples are effectively synchronized in the respective cell cycle phase. (**C**) The number of mitotic cells was scored by microscopy based on chromosome condensation (DNA stained with DAPI). Graph is average of 3 independent experiments with ~50 cells scored per experiment. Error bars represent standard error of the mean (SEM). (**D**) RPE-1 cells were synchronized as described in (**A**). Before harvesting, cells were incubated with S35-methionine for 10 min to radioactively label newly synthesized proteins. The left panel shows the autoradiograph of newly synthesized proteins. The middle panel shows total protein content of the cells stained by Coomassie. Right panel shows quantification of autoradiographs of 3 independent experiments, normalized to total protein. Mean and standard deviation (SD) are shown.

washed out and the cells progress into M and then G1, and pure (95%) populations of M and G1 cells can be obtained based upon the timing of release from G2 arrest (*Figure 1B,C*). Thus, this cell synchronization protocol is minimally perturbing and avoids the use of drugs that arrest cells in mitosis by targeting microtubules, which are known to affect translation (*Sivan et al., 2011*; *Coldwell et al., 2013*).

Using S35-methionine labeling, we found a global decrease in mRNA translation during mitosis when compared to G2 and G1 cells respectively (*Figure 1D*). This finding confirms the global reduction in translation during mitosis reported by other studies (*Fan and Penman, 1970*; *Bonneau and Sonenberg, 1987*; *Pyronnet et al., 2001*); however, the magnitude of the effect seen here is smaller (~35% reduction in our study vs 70% found previously [*Fan and Penman, 1970*]). This difference might be due to the use of microtubule inhibitors in cells arrested in mitosis in previous studies, which can cause global translational repression (*Sivan et al., 2011*; *Coldwell et al., 2013*).

Next, we analyzed genome-wide mRNA levels in G1, G2 and M phase cells by deep sequencing (*Supplementary file 2*). Comparing mRNA abundance in G2 vs M cells, we detected only 25 genes that were down-regulated in M and not a single up-regulated gene (>threefold, see 'Materials and methods' section) (*Figure 2A*, red bars, left graph). In contrast, when G2 cells were compared to G1 cells, 220 genes were down-regulated and 82 genes up-regulated in G1 (*Figure 2A*, red bars, middle graph). Similarly, when M cells were compared to G1 cells, many mRNAs were changed (138 up and 156 down in M) (*Figure 2A*, red bars, right graph). The mRNAs that changed between G1 and G2 were mostly the same set that also changed between G1 and M (*Figure 2—figure supplement 1A*); 96% of mRNAs that are up-regulated in G1 vs G2 are also increased at least twofold in G1 vs M. Similarly, 80% of mRNAs that are down-regulated in G1 vs G2, are also decreased at least twofold in G1 vs M. Together, these results indicate that the mRNA content of G2 and M phase cells is very similar, but distinct from G1 phase cells.

Next, we subjected ribosome FPs to deep sequencing to examine whether individual mRNAs are differentially translationally regulated at different stages of the cell cycle (*Supplementary file 3*). We refer to this regulation as *gene-specific* translational to distinguish it from the global translational repression described above. The number of ribosome FPs (which reports on the amount of total translation) was determined for each mRNA and was divided by the total mRNA abundance to obtain the TE. The vast majority of gene-specific changes in TE were observed when M phase transcripts were compared with either G2 or G1; 199 and 92 genes were translationally regulated between M and either G2 or G1, respectively. In contrast, only 13 genes showed changes in translation between G2 and G1 (*Figure 2A*, blue bars; transcripts with >threefold difference in TE, and >twofold difference in ribosome footprint (FP) density were scored as translationally controlled, see 'Materials and methods' for more details). Thus, in contrast to mRNA abundance, which is similar in G2 and M, but distinct in G1, TE is similar in G2 and G1, but very different in M.

When we analyzed mRNA abundance of the 199 genes that showed gene-specific regulation in M, we found that their mRNA levels were largely constant throughout the cell cycle (*Figure 2B*). Similarly, the TE of genes known to be transcriptionally regulated was largely constant (*Figure 2C*). These results indicate that gene-specific translational regulation affects a different set of genes than transcriptional regulation.

The vast majority of the 199 mRNAs that show *gene-specific* translational regulation in M compared to G2 were repressed rather than activated; comparing M to G2, 182 were translationally downregulated in M and only 17 were upregulated (*Figure 2A*, blue bars, middle graph; *Figure 2—figure supplement 1B*). Similarly, of the 92 mRNAs that translationally regulated between M and G1, 86 were repressed in M, and only 6 were activated (*Figure 2A*, blue bars, right graph; *Figure 2—figure supplement 1B*). To test whether the same set of mRNAs that was translationally repressed at mitotic entry were de-repressed at mitotic exit, we compared the overlap in mRNAs repressed in M vs G2 and M vs G1. The genes that were translationally repressed in M vs G2 were mostly also repressed in M vs G1; of the 182 genes that were repressed in M compared to G1, 87% were repressed >twofold in M compared to G1. Furthermore, there is a good correlation in the fold change in TE between G2 vs M and G1 vs M for individual mRNAs (*Figure 2D*). In summary, when cells progress from G2 to M, gene-specific translational regulation is dominated by repression, and the genes that are translationally repressed as cells enter mitosis are mostly re-activated upon mitotic exit.

It is important to note that fold change values noted above are relative to the average mRNA of the biological sample (as ribosome profiling only reports on relative changes). Thus, specific mRNAs that are translationally repressed threefold relative to other mRNAs in mitosis, are repressed ~fourfold relative to the same gene in G2 phase (given the global ~35% translational repression that acts on all mRNAs during mitosis). Similarly, the small number of mRNAs that are translationally activated by threefold in mitosis, are only expressed ~twofold higher than in G2 phase. Thus, we conclude that the vast majority of mRNAs that undergo gene-specific regulation are translationally repressed in mitosis.

Next, we examined whether there were particularly types of genes that were predominantly regulated by translational vs transcriptional control, so we performed gene ontology enrichment analysis using the functional annotation tool DAVID (*Huang da et al., 2009*). Many genes that exhibited variations in mRNA levels during the cell cycle are involved in cell division (p-values $< 10^{-9}$, see 'Materials and methods'). In contrast, the translationally regulated genes were functionally very different from the transcriptionally regulated genes and included many signaling molecules, transcription factors, and transmembrane proteins (significantly enriched with p-values $< 10^{-8}$)

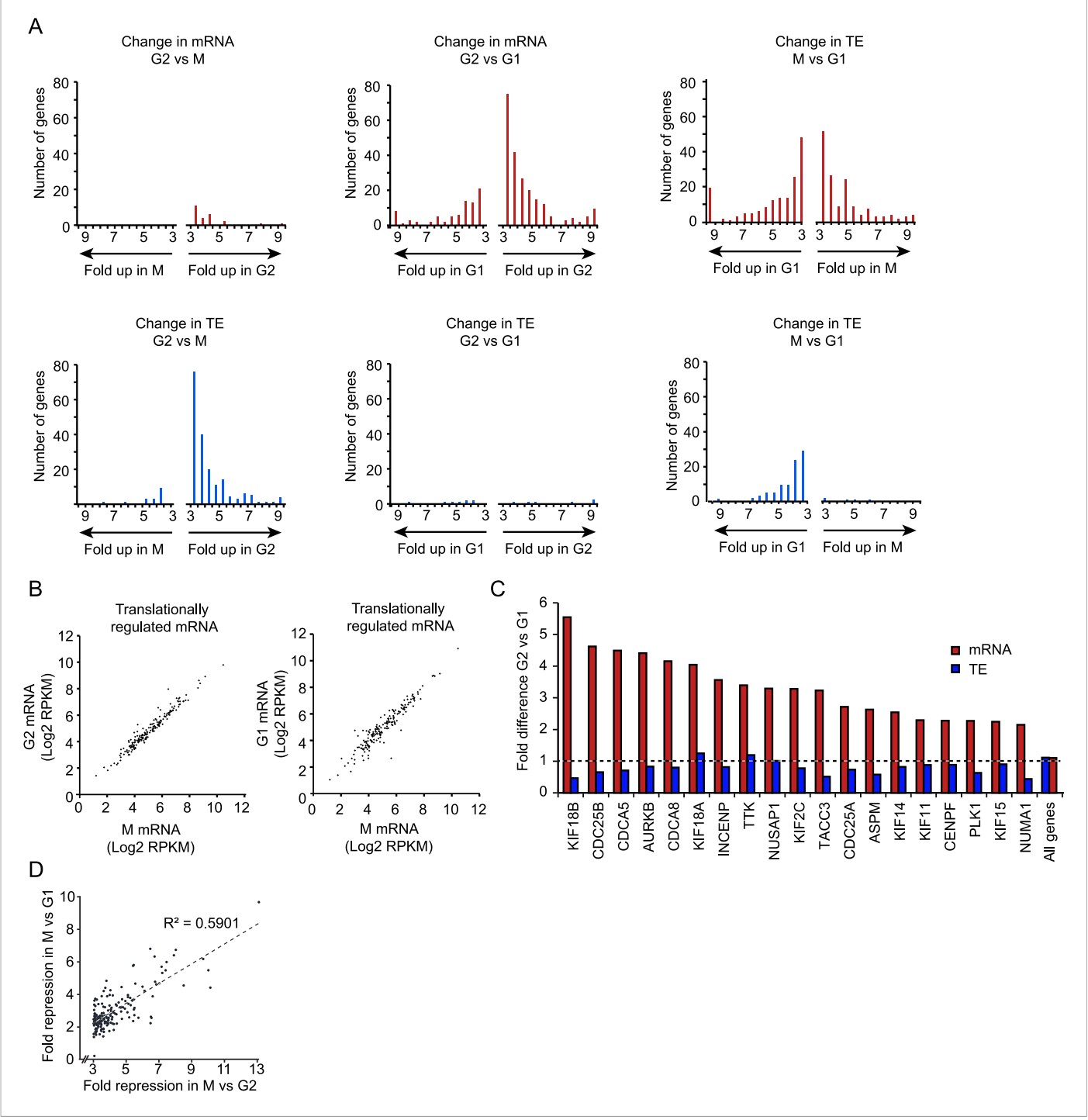

**Figure 2**. Transcriptional and translational regulation affect distinct cell cycle transitions. (**A**) For each gene the ratio of mRNA levels (red bars) and translation efficiency (TE) (blue bars) was determined for G2 vs M (left) or G2 vs G1 (middle) and M vs G1 (right). The number of genes that showed changes in mRNA levels (>threefold difference, red bars) or changes in TE (or >threefold difference in TE combined with >twofold difference in the ribosome footprint value, blue bars) was plotted in a pair of histograms. (**B**) mRNA levels are plotted for all genes that are translationally regulated in M vs G2. The left graph shows the mRNA levels in M compared to G2, the right graph shows mRNA levels of M compared to G1. Note that mRNA levels of translationally regulated mRNAs are similar in G2, M and G1. (**C**) 19 well characterized cell cycle proteins that show strong cell cycle-dependent regulation of mRNA levels were manually selected. Fold difference between G2 and G1 in mRNA levels (red bars) and TE (blue bars) is shown. While there is a large change in mRNA levels, the TE is similar in G2 and G1 for the majority of these mRNAs. (**D**) The subset of genes that was translationally repressed in M compared to G2 (182 genes) was selected and the fold difference in TE for M vs G2 was plotted against the fold difference in TE for M vs G1. Results show

*Figure 2. continued on next page*

Figure 2. Continued

that genes which are translationally repressed in M compared to G2, or repressed to similar levels in M when compared to G1.

The following figure supplements are available for figure 2:

**Figure supplement 1**. Transcriptional and translational regulation affect distinct cell cycle transitions.

**Figure supplement 2**. Translational regulation affects many cell cycle-dependent processes.

**Figure supplement 3**. Excessive PP1γ and PP2aβ activity perturbs chromosome segregation.

(see *Figure 2—figure supplement 2*). Manual curation of the mRNAs that showed gene-specific translational repression during mitosis revealed regulation of several components of the same pathway. For example, multiple components of the PI3 kinase pathway were translationally repressed during mitosis (*Figure 2—figure supplement 2A*). We also found mitosis-specific translational downregulation of the mitotic phosphatases PP1γ and PP2aβ (*Figure 2—figure supplement 2B*). During mitosis, these phosphatases are strongly inhibited through phosphorylation and binding to inhibitory proteins (*Wurzenberger and Gerlich, 2011*), suggesting that mitotic translational repression may represent an additional back-up mechanism to inhibit protein function (see Discussion). Consistent with this notion, we found that overexpression of either PP1γ or PP2aβ phosphatase strongly disrupted normal cell division (*Figure 2—figure supplement 3A–C*). We also found a strong translational repression of two key regulators of centriole duplication, Plk4 and CP110 (*Figure 2—figure supplement 2C*) (*Chen et al., 2002*; *Habedanck et al., 2005*). Finally, the majority of histones showed a strong reduction in protein synthesis in M compared with G2. Newly synthesized histones may not incorporate readily into highly condensed mitotic chromosomes, which perhaps could explain why their translation is reduced during mitosis, although we cannot completely rule out a small contamination of S-phase cells in the G2 sample which might give rise to an apparent high level of translation of histone mRNAs in G2. We also found a few mRNAs that were translationally increased in mitosis as compared to both G1 and G2, although most were below our threshold of threefold change, indicating the changes were subtle. Included in this list are genes involved in cytoskeleton function and DNA replication initiation (*Figure 2—figure supplement 2E,F*), the latter of which may reflect the ability of cells to license DNA replication at the end of mitosis (*Clijsters et al., 2013*). Taken together, these results show that transcriptional and gene-specific translational control dominate at different stages of the cell cycle (transcription at the G1-to-G2 transition and translational regulation dominating at the more rapid G2-to-M transition) and regulate a distinct set of genes.

## A live cell fluorescence reporter for translation

To validate the results obtained in our ribosome profiling experiments in living cells, we developed a fluorescence-based reporter to analyze translation of individual transcripts in living cells. In brief, the reporter consists of a GFP fused to an inducible degron, which is continuously degraded until a small molecule stabilizer is added (*Iwamoto et al., 2010*) (*Figure 3A*). A mCherry protein expressed independently from the same transcript was used for normalization (*Figure 3—figure supplement 1A,B*). Our assay is similar to another recently developed method for measuring translation rates in cells (*Han et al., 2014*), although our system included an mCherry protein that was expressed from the same mRNA using a P2A site. Upon addition of the stabilizer drug, time-lapse microscopy revealed an increase in the GFP/mCherry ratio over time, which reflects the translation rate of the GFP (*Figure 3B,C*, see 'Materials and methods').

To test for translational regulation, the 5′ and 3′UTRs of the control reporter were replaced by the 5′ and 3′ UTRs of genes that were identified by ribosomal profiling as translationally repressed in mitosis; UTRs of two genes that were not translationally regulated were tested as controls. For these experiments, we used the UTRs of mRNA isoforms that closely matched the isoforms expressed in RPE1 cells, as determined by our RNA-seq data (*Figure 3—figure supplement 1C,D*) While the UTRs of control transcripts did not lead to translational regulated of the GFP reporter, time-lapse imaging

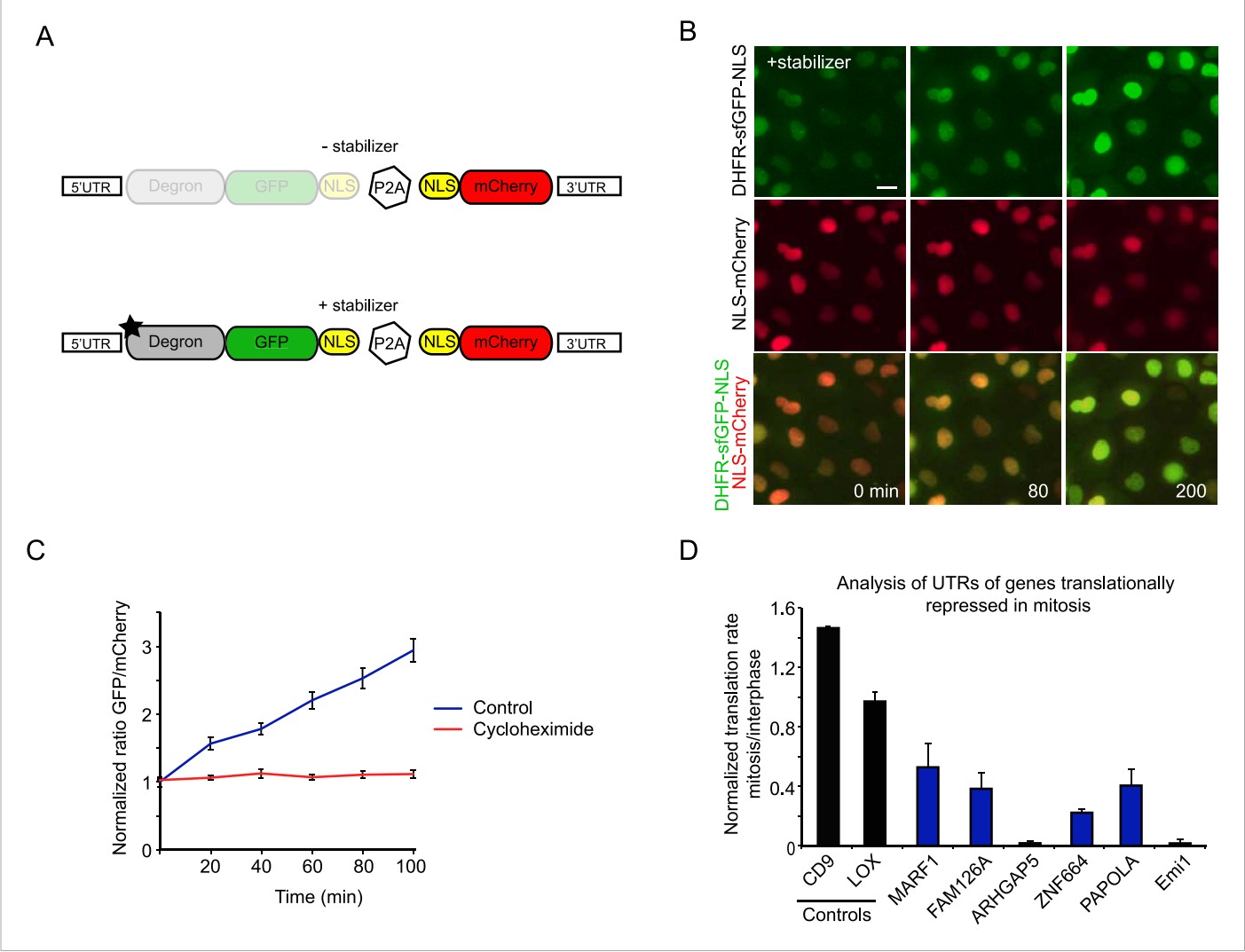

**Figure 3**. Analysis of translation efficiency in living cells using a fluorescence-based translation reporter. (**A**) Schematic representation of the live-cell translation reporter. An inducible degron (DHFR-Y100I) fused to sfGFP and an NLS is separated from an NLS-mCherry protein by a P2A ribosome skipping sequence, which allows these two proteins to be synthesized as separate proteins from a single transcript. Upon addition of the small molecule stabilizer trimethoprim (TMP), newly synthesized DHFR-sfGFP-NLS is stabilized and GFP fluorescence increases over time due to new GFP protein synthesis. Thus, GFP fluorescence increase reports on translation efficiency. The mCherry signal is used to normalize for the plasmid copy number per cell. (**B**) RPE-1 cells stably expressing the reporter were treated with 50 μM TMP and followed by time-lapse microscopy. Scale bar, 20 μm. Time is indicated in min. (**C**) Quantification of GFP/Cherry ratio (mean and standard deviation, n = 8 cells) with or without cycloheximide treatment. (**D**) 5′ and 3′ UTRs from indicated genes were inserted in the reporter. Cells expressing the different reporters were blocked in mitosis with taxol, treated with TMP and imaged for 4 hr. To determine the translation rate, the GFP/mCherry ratio was calculated at the start and end of each video for both interphase and mitotic cells. The ratio of translation rates in mitosis and interphase for each reporter is shown. For Emi1, mitotic cells were compared with G2 phase cells only, as translation was also reduced in G1 (unpublished observation). Results are mean and SEM of 3 independent experiments with 10–20 cells analyzed per condition per experiment.

The following figure supplement is available for figure 3:

**Figure supplement 1**. Analysis of mRNA sequences that confer translational regulation.

revealed that the transcripts with UTRs from translationally repressed mRNAs showed decreased synthesis in mitosis compared with interphase (*Figure 3D*). These results provide strong, independent validation of the ribosome profiling dataset and show that regulatory elements present in 5′ and 3′ UTRs confer translational repression in mitosis.

## The role of translational repression of Emi1

To study the function of translational repression in mitosis, we focused on Emi1, a highly repressed gene identified in our ribosome profiling dataset (*Supplementary file 3*) and translational reporter assay. (*Figure 3D*). First we wished to confirm that regulation of Emi1 expression is due solely to translational regulation, not to changes in Emi1 mRNA expression. To establish that mRNA levels of Emi1 remain unchanged as cells progress into mitosis, we performed single molecule FISH and found that Emi1 mRNA was present at similar levels in G2 and M (*Figure 3—figure supplement 1E*). However, since these FISH probes detect all known Emi1 isoforms, we also wished to determine whether isoform specific regulation may occur. First, we confirmed that the same mRNA isoforms were expressed in G2 and M (*Figure 3—figure supplement 1D*). Furthermore, we mapped the regulatory element in the Emi1 UTRs, and found that the translational repression was conferred by the 3′UTR of Emi1 alone, which is identical in all Emi1 mRNA isoforms (*Figure 3—figure supplement 1D,F*). Consistent with this, all 5′UTR isoforms in combination with the Emi1 3′UTR were regulated in a similar manner (*Figure 3—figure supplement 1D,F*). Together, these results confirm the conclusion that the inhibition of Emi1 protein synthesis in M is due to translational repression, and show that the 3′UTR of Emi1 is sufficient for this translational regulation.

Emi1 is a potent inhibitor of the APC bound to its activator Cdh1 (APC/Cdh1). In G2 phase, Emi1 inhibits APC/Cdh1, and this inhibition is required to allow accumulation of cyclins in G2 and for mitotic entry (*Reimann et al., 2001*; *Hsu et al., 2002*; *Di Fiore and Pines, 2007*). However, in early mitosis, Emi1 protein is inactivated both through protein degradation and CDK1-dependent phosphorylation (*Reimann et al., 2001*; *Guardavaccaro et al., 2003*; *Margottin-Goguet et al., 2003*; *Moshe et al., 2011*), which is likely required for APC/Cdh1 activation and APC substrate degradation in the subsequent telophase/G1. Given these post-translational mechanisms for inhibiting Emi1, it is unclear whether translational repression of Emi1 also serves a role in suppressing its activity during mitosis.

To test the importance of translational inhibition of Emi1 in mitosis, we expressed mCherry-Emi1 (a functional fusion protein [*Di Fiore and Pines, 2007*]) from a plasmid lacking native UTRs and thus lacking translational regulation, and examined APC/Cdh1 activation as cells progressed through mitosis into the next G1 phase. First, we confirmed that mCherry-Emi1 was degraded in late G2/early mitosis (*Figure 4—figure supplement 1A*), as found previously (*Guardavaccaro et al., 2003*; *Margottin-Goguet et al., 2003*). As a readout of APC/Cdh1 activity, we expressed fluorescently-tagged Aurora A, Plk1 or CDC20, all of which are APC/Cdh1 substrates. (*Peters, 2006*). These fluorescence reporter substrates were all rapidly degraded in anaphase/telophase in control cells as expected (*Figure 4A–C*). When Emi1 lacking its normal 5′ and 3″ translation regulatory elements was expressed, cells progressed through mitosis normally and the bulk of Emi1 protein was degraded; however, APC/Cdh1 activation in telophase was partially inhibited, as indicated by the decreased degradation of APC/Cdh1 substrates (*Figure 4A–C* and *Video 1*). Importantly, exogenous Emi1 was expressed at similar levels as the endogenous gene in these experiments, as determined by transcript counting by single molecule FISH (*Figure 4—figure supplement 1B*). These results show that Emi1 can inhibit APC/Cdh1 activation at the end of mitosis and suggest that Emi1 protein degradation is insufficient to completely inhibit Emi1 activity in the presence of continued Emi1 synthesis.

To test whether translational repression of Emi1 would allow APC activation under these conditions, we replaced the control UTRs of mCherry-Emi1 with Emi1's native UTRs and examined APC activation at mitotic exit. Strikingly, translational repression of Emi1 in mitosis allowed a substantially higher degree of APC/Cdh1 activation at comparable Emi1 expression levels (*Figure 4D,E* and *Figure 4—figure supplement 1C*). To confirm that the UTRs of Emi1 were enhancing APC/Cdh1 activation solely through mitosis-specific translational repression, rather than through an alternative mechanism, for example by stimulating specific mRNA localization, we sought to confer mitosis-specific translational repression on Emi1 mRNA independently of the Emi1 UTRs. To this end, we replaced the Emi1 UTRs with the UTRs of an unrelated mRNA, ARHGAP5, which have a completely different sequence, but confer translational repression in mitosis to a similar extent as Emi1 UTRs (*Figure 3D*). Indeed, mitosis-specific translational repression of mCherry-Emi1 by ARHGAP5 UTRs allowed APC activation at the end of mitosis to a similar extent as Emi1 UTRs (*Figure 4D,E*). Furthermore, examination of Emi1 mRNA localization during mitosis did not reveal localization to a specific site in the cell (*Figure 4—figure supplement 1D*), arguing against a role for the UTRs of Emi1 in mRNA localization, but rather indicating that their primary function is to repress translation during mitosis.

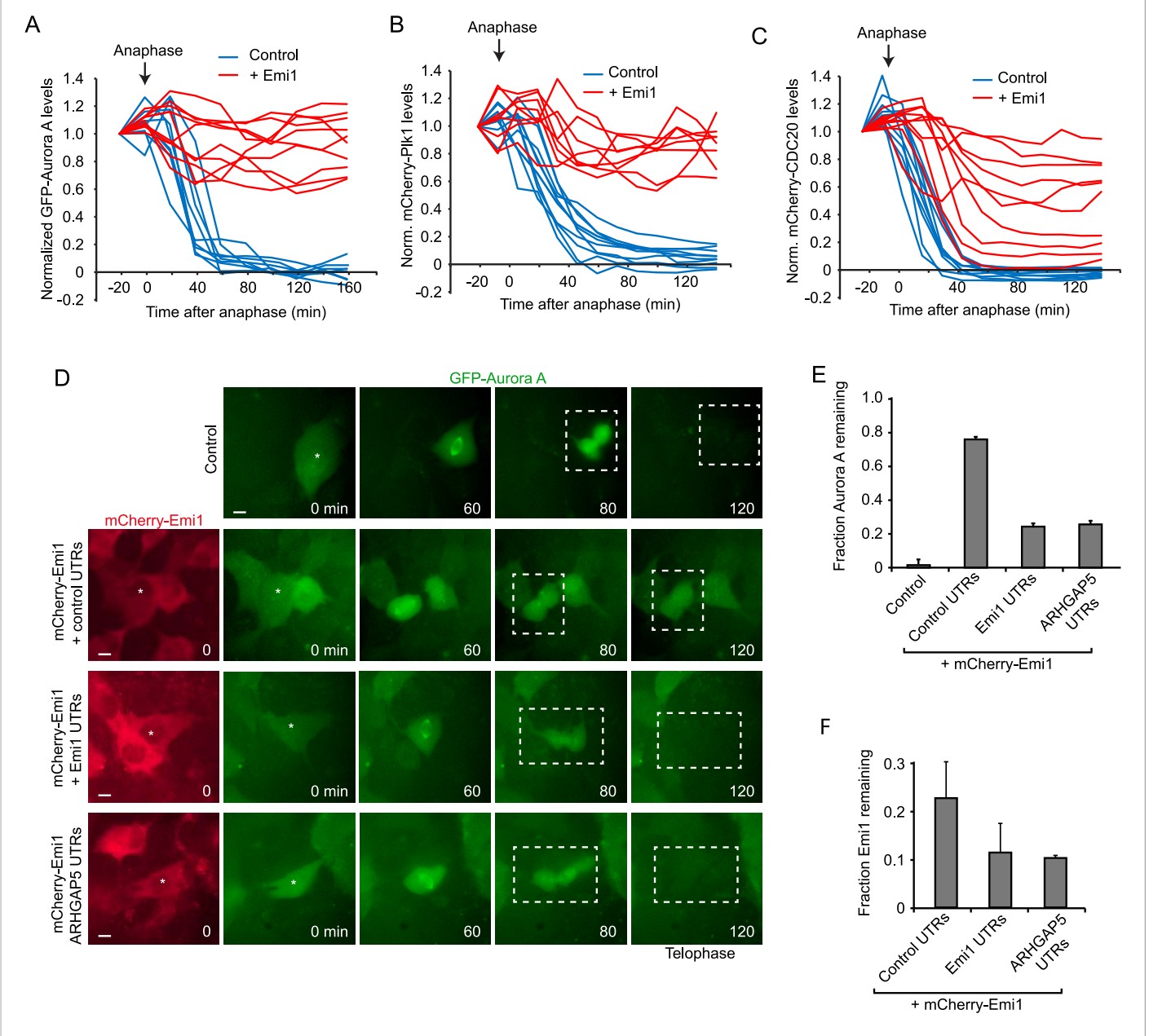

**Figure 4**. Translational inhibition of Emi1 during mitosis promotes APC/Cdh1 activation in telophase. (**A–C**) RPE-1 cells expressing Aurora A-GFP (**A**), mCherry-Plk1 (**B**) or mCherry-CDC20 (**C**) alone or combined with fluorescently tagged Emi1 were analyzed by time-lapse microscopy and the protein degradation rates were assayed through quantification of fluorescence intensities over time as cells progressed through mitosis. (**D–F**) RPE-1 cells stably expressing Aurora-GFP and, where indicated, mCherry-Emi1 with indicated UTRs, were analyzed be time-lapse microscopy. Representative images (**D**) and quantification of Aurora A-GFP levels (**E**) and mCherry-Emi1 levels (**F**) are shown. Asterisks and dotted boxes mark dividing cells. Degradation of Aurora A normally occurs between anaphase onset and telophase. For quantification only cells were included that had very low mCherry-Emi1 fluorescence to ensure low expression level of exogenous Emi1 (see also *Figure4—figure supplement 1C,D*). Scale bar, 10 µm. Mean and standard error of 3 independent experiments, with ~10 cells analyzed per experimental condition per experiment.

The following figure supplement is available for figure 4:

**Figure supplement 1**. Translational repression of Emi1 by its UTRs facilitates APC activation.

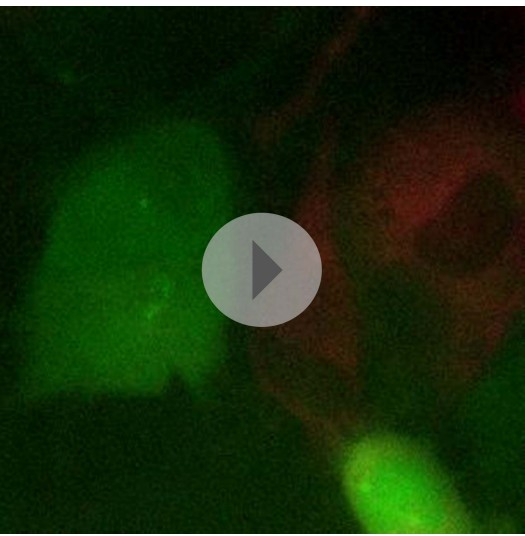

**Video 1.** Video of two RPE-1 cells going through cell division. The first cell to divide expresses both Aurora A-GFP and mCherry-Emi1, while the second cell only expresses Aurora A-GFP Note that upon division, Aurora A-GFP degradation is perturbed in the cell expressing mCherry-Emi1. Time interval between images is 20 min.

To understand how translational repression of Emi1 contributes to APC activation, we examined Emi1 protein levels during mitosis in the presence and absence of translation repression. Interestingly, Emi1 was more completely depleted when protein degradation was combined with translational repression (*Figure 4F*). These results demonstrate that inactivation of existing Emi1 protein, through degradation and potentially inhibitory phosphorylation (*Moshe et al., 2011*), is insufficient to completely inhibit Emi1 activity, but when post-translational inactivation is combined with translational repression, Emi1 is more thoroughly inhibited, which allows robust APC activation.

## Discussion

### Evidence supporting gene-specific translational regulation during mitosis

Previous work had found a global translational repression during mitosis (*Fan and Penman, 1970*). Here we confirm a modest (35%) and general repression of translation in mitosis, but in addition, our genome-wide analyses revealed much larger (>300%) effects on TE of several hundred specific mRNAs, the large majority of which involve translational repression rather than activation. Thus in summary, the mitotic translational program is dominated by a modest global translational repression, combined with potent repression of a subset of mRNAs, which acts on top of the global repression. Our follow-up studies on Emi1 suggests that its translational repression acts as a mechanism to enhance post-translational protein inactivation, which could represent a more general function of translational repression in mitosis and in other processes.

The evidence for mitosis-specific translational repression of specific mRNAs is supported by two separate methodologies. The first evidence comes from ribosomal profiling which measures ribosome occupancy of native transcripts, but does not provide a direct measure of translation itself. The second method involves the use of an introduced (thus non-native) reporter mRNA that provides a real-time, live-cell readout of protein production from translation. In the test cases where both methods were applied, we find good agreement in the results, providing strong support for widespread gene-specific translational repression in mitosis.

In this work, we avoided microtubule depolymerizing drugs for cell cycle synchronization, as employed by many other studies of translation in mitosis, since microtubule depolymerization has been recently shown to have a profound effect on mRNA translation (*Coldwell et al., 2013*). However, one possible concern is whether the CDK1 inhibitor RO-3306 used here to arrest cells in G2 might influence translation as well and thus affect the conclusions of this study. CDK1, for example, is known to phosphorylate a number of translation factors (*Heesom et al., 2001*; *Dobrikov et al., 2014*; *Shuda et al., 2015*) and this could affect TE. However, we feel that the major conclusions regarding selective translational repression in mitosis are unlikely to be due to the use of this CDK1 inhibitor for cell synchronization for three reasons. First, the genome-wide comparison of TE of G1 vs M cells (neither of which have any drug present) yields very similar results to the comparison of G2 (RO-3306 present) vs M cells. Second, we confirmed translational regulation of a selected subset of mRNAs using a single cell fluorescence-based assay that does not involve the use of the CDK1 inhibitor. Third, the CDK1 inhibitor is only present in our G2 sample, a time at which CDK1 is mostly inactive in unperturbed cells (*Jackman et al., 2003*; *Gavet and Pines, 2010*). Consistent with this, the translation inhibitor 4E-BP1, which is phosphorylated by CDK1 in mitosis, is not detectably phosphorylated in G2 (*Heesom et al., 2001*). Thus, our synchronized G2 sample closely mimics the G2 phase of unsynchronized cells with

respect to CDK1 activity. Together, these results strongly support the conclusion that the observed translational effects reported in this study are due to distinct cell cycle states and not an effect of drug treatment.

## Use of transcriptional and translational regulation for distinct cell cycle transitions

Interestingly, the sets of genes that show altered TE and mRNA abundance (likely due, at least in part, to transcriptional regulation) are largely non-overlapping (*Figure 2B,C*). Furthermore, the cell cycle timing of transcriptional and translational regulation also differ; the majority of gene-specific translational changes occur at entry and exit from mitosis (G2-to-M and M-to-G1 transitions), while transcriptional changes predominate between G1 and G2. We also show that the vast majority of the ~200 translationally-regulated mRNAs are repressed (several fold below the modest global down-regulation of translation in mitosis) rather than activated. Thus, we conclude that the dominant function of gene-specific translational regulation during mitosis is to potently inhibit synthesis of a relatively small subset of the proteome. Our work complements another cell cycle ribosomal profiling study by Stumpf et al. (*Stumpf et al., 2013*), which focused on translational changes associated with S phase, and together these studies provide a complete overview of translational regulation during the cell cycle.

The distinct use of transcriptional and translational regulatory mechanisms is consistent with the timing of the cell cycle transitions for which they are used; regulating transcription is a relatively slow mechanism for altering protein synthesis rates, because of the time required for transcription, mRNA processing/nuclear export and mRNA turnover. This limits the usefulness of transcriptional regulation to longer time transitions in the cell cycle, specifically from G1 to S or G2. On the other hand, translational control alters protein synthesis rates almost instantaneously, which makes it well suited to the short time scale of mitosis, which is generally less than 1 hr in most somatic cells with doubling times of 1–2 days. Interestingly, a recent study of mouse dendritic cell activation, which happens at a time-scale of many hours, similar to the G1-S transition, found that the vast majority of changes in protein synthesis were due to altered mRNA abundance, rather than due to changes in translation rates (*Jovanovic et al., 2015*). A second unique feature of translational control is its rapid reversibility, enabling protein synthesis to restart quickly when cells exit from mitosis and enter G1. Thus, translational control may be employed when acute changes in protein synthesis are needed (e.g., entry into and exit from mitosis), while transcriptional control is used to affect slower changes in gene expression.

## Gene-specific mitotic translational regulation is mainly used to repress, not activate translation

A surprising finding of this study is that very few mRNAs showed substantial (>threefold), gene-specific translational activation during mitosis. In previous reported cases of translational regulation of specific mRNAs during mitosis, the findings were predominantly of translational activation through an IRES-dependent mechanism (*Cornelis et al., 2000*; *Pyronnet et al., 2000*; *Qin and Sarnow, 2004*; *Wilker et al., 2007*; *Marash et al., 2008*; *Ramirez-Valle et al., 2010*). While we also found a small number of mRNAs that were translated more efficiently during mitosis (16 and 6 mRNAs were translationally upregulated >threefold in mitosis compared to G2 and G1, respectively), the vast majority of regulated mRNAs were translationally repressed (187 and 97 mRNAs were down-regulated in mitosis compared to G2 and G1, respectively). One possible explanation for this discrepancy is that the IRES-mediated upregulation of translation in mitosis may be relatively weak compared to the cut-off (>threefold) that we have used in this study to identify changes in gene-specific TE. A previous study of IRES-mediated translational upregulation found that only one of three proteins examined was upregulated by > threefold (*Qin and Sarnow, 2004*). Furthermore, a recent study found that translation in mitosis is dominantly cap-dependent (*Shuda et al., 2015*), suggesting that IRES-mediated translation may only make a minor contribution to the translational landscape in mitosis. The results from our work and Shuda et al., while not ruling out IRES-dependent translational activation of some genes in mitosis, indicates that the dominant function of gene-specific translational regulation is to shut down rather than activate synthesis of proteins during mitosis.

Our results, along with others, also reveal a number of interesting differences when comparing mitosis in somatic cells to studies of translational regulation during meiosis and early embryonic development (*Mendez and Richter, 2001*; *Groisman et al., 2002*). Oocytes stockpile maternal mRNA, as new transcription does not occur until the mid-blastula stage of development. Due to the absence of transcriptional control, translational regulation is the main mechanism by which protein synthesis rates are tuned. In these systems, translational regulation occurs largely by modulating poly (A) tail length (*Weill et al., 2012*; *Subtelny et al., 2014*). While changes in poly(A) tail length also occur on a subset of mRNAs during the mitotic cell cycle (*Novoa et al., 2010*), regulation of poly(A) tail length does not appear to be a general mechanism for controlling TE in somatic cells (*Subtelny et al., 2014*). Thus, translational control in meiosis is most likely mechanistically distinct from the gene-specific translational regulation during mitosis described here. Consistent with this, we find that the large majority of regulated mRNAs are translationally repressed during mitosis, while in meiosis regulatory mechanisms mainly function to specifically activate a subset of mRNAs (*Mendez and Richter, 2001*). Furthermore, the sets of genes that are regulated during meiosis and mitosis are largely non-overlapping. Interestingly, Emi1 is an exception to this rule, as it is translationally regulated during both meiosis (*Belloc and Mendez, 2008*) and mitosis (this study). However, in meiosis it is translationally activated through control of its poly(A) tail length by the RNA binding protein CPEB (*Belloc and Mendez, 2008*). This does not appear to be the case during mitosis, as mutation of all CPEB binding sites in the 3′UTR of Emi1 does not prevent translational repression in mitosis (unpublished observation). Therefore, we conclude that the gene-specific translational control program in mitosis identified here is distinct from the meiotic translation program.

## Translational repression as a mechanism to enhance post-translational protein inactivation

What might be the function of translational repression during mitosis? The average protein half-life is many hours (*Schwanhausser et al., 2011*), while mitosis takes less than an hour, so inhibition of protein synthesis for this short period of time is not expected to have a major impact on overall protein levels. Our analysis of Emi1 offers a clue to the role of gene-specific translational repression during mitosis. During mitosis pre-existing Emi1 protein is inactivated through phosphorylation and ubiquitin-mediated degradation (*Guardavaccaro et al., 2003*; *Margottin-Goguet et al., 2003*; *Moshe et al., 2011*). However, we find that in the presence of continued protein synthesis, degradation cannot remove all of the Emi1, resulting in a small amount of residual Emi1 protein levels at the end of mitosis, which can interfere with full APC activation. In contrast, when new Emi1 protein synthesis is inhibited through translational repression, Emi1 is eliminated more completely (*Figure 5*). Transcriptional inhibition at the G2/M transition would not afford a similar effect, since protein synthesis rates would not decline substantially within the required time-scale due to relatively slow mRNA turnover.

Our model for the inhibition of Emi1 during mitosis involves the synergistic effects of protein degradation along with the inhibition of protein synthesis via translational repression. However, of the proteins that are translationally repressed at the G2-to-M transition, very few are known to be degraded during mitosis, leaving open the question of why they are translationally repressed. One possibility is that these proteins are inactivated during mitosis through other post-translational mechanisms, such as phosphorylation. As newly synthesized proteins are unphosphorylated and thus active, inhibition of translation will limit the formation of such new, active protein and thus enhance post-translational protein inactivation.

In summary, our work provides a genome-wide view of translationally controlled mRNAs in mitosis and provides a new hypothesis for the role of translational repression during mitosis; as a mechanism to augment post-translational protein inactivation.

## Materials and methods

### Ribosome profiling and mRNA sample preparation

For ribosome profiling, cells were treated with cycloheximide before lysis for 2 min as previously described, which does not substantially alter overall mRNA read density (*Ingolia et al., 2011*). Cells were lysed in lysis buffer (20 mM Tris pH 7.5, 150 mM KCl, 5 mM MgCl$_2$, 1 mM dithiothreitol, 8% glycerol) supplemented with 0.5% Triton X-100, 30 U/ml Turbo DNase (Ambion, Life Technologies,

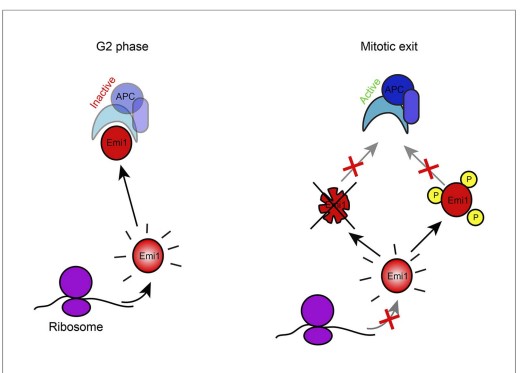

**Figure 5**. Model of Emi1 regulation in mitosis. Model of Emi1 regulation in G2 and M. In G2, new Emi1 protein is continuously synthesized and existing Emi1 protein is stable allowing robust buildup of Emi1 protein levels and inhibition of APC/Cdh1 (left). In mitosis, Emi1 synthesis is repressed and pre-existing Emi1 protein is inactivated through protein degradation and CDK1-dependent phosphorylation, resulting in full inhibition of Emi1 activity, allowing APC/Cdh1 activation in telophase (middle) (Note- additional Emi1-independent mechanisms (not depicted) keep APC/Cdh1 inactive in (pro)metaphase).

CA, United States) and 100 µg/ml cycloheximide (Sigma Aldrich, MO, United States); ribosome-protected fragments were then isolated and sequenced as previously described (*Ingolia et al., 2011*). Total RNA was isolated from cells using Trizol Reagent (Ambion). Polyadenylated RNA was purified from total RNA using magnetic oligo(dT) beads. The resulting mRNA was modestly fragmented by partial hydrolysis in bicarbonate buffer so that the average size of fragments was ~80 bp. The fragmented mRNA was separated by denaturing PAGE and fragments 50–80 nt were selected. The sequencing libraries were prepared and sequenced as previously described (*Ingolia et al., 2011*).

## Sequence alignments and normalization

Prior to alignment, linker and poly(A) sequences were removed from the 3′ ends of reads. Bowtie v0.12.7 (*Langmead et al., 2009*) (allowing up to 2 mismatches) was used to perform the alignments. First, reads that aligned to human rRNA sequences were discarded. All remaining reads were aligned to the human (hg19) genome. Finally, still-unaligned reads were aligned to known canonical mRNA. Reads with unique alignments were used to compute the total number of reads obtained for each transcript. FP alignments were assigned to specific P site nucleotides by using the position and total length of each alignment, calibrated from FPs at the beginning and the end of CDSes. FP and mRNA densities were calculated in units of reads per kilobase per million (RPKM) in order to normalize for gene length and total number of reads per sequencing run. The density of ribosome FPs is used as a measure of the rate of translation and TE is defined as the ratio of FP density/total mRNA. For each sample, two biological replicates were generated and the sum of the number of reads for each gene over the two replicates was calculated. We found that genes that had more than 200 reads total showed very strong reproducibility between replicates (*Supplementary file 1A*). Therefore, we excluded from the analysis genes that had less than 200 reads. In addition, genes for which the RPKM value of the two biological replicates showed a difference of more than threefold, were excluded from further analysis. For genes that passed both filters, the RPKM values of the two biological replicates were averaged. $R^2$ values and standard deviations for the ratio of the replicates are listed in *Supplementary file 1*. For TE analysis, only genes for which mRNA samples had at least 200 reads were included. Unless stated otherwise, we used a cutoff of threefold difference between samples. For differences in TE, we used a >threefold difference in TE combined with a >twofold difference in FP RPKM. We reasoned that an increase in TE is only biologically meaningful if it is accompanied by an increase in FP RPKM, as this indicates that the total amount of protein synthesis is increased on the specific mRNA. A >threefold difference cutoff was >sixfold the standard deviation between replicates of all samples and was therefore highly significant. Functional classification of genes was done using the online webserver DAVID (*Huang da et al., 2009*). The data described in this study have been deposited in NCBI's Gene Expression Omnibus (GEO) and are accessible through GEO Series accession number GSE67902 (http://www.ncbi.nlm.nih.gov/geo/query/acc.cgi?acc=GSE67902).

## Cell culture, generation of stable cell lines and drug treatment

RPE-1 cells were grown in DMEM:F-12 medium supplemented with 10% FCS and antibiotics. For all stable cell lines, lentiviruses were made in 293 cells using the pHR vector and separate packaging vectors. For transduction, RPE-1 cells were incubated with virus for 24 hr. RO-3306

(Axon medchem, The Netherlands) was dissolved in DMSO and used at 6 µM for cell synchronization. CHX (Sigma) was dissolved in ethanol and used at 100 µg/ml. TMP (Sigma) was dissolved in DMSO and was used at 50 µM.

## Time lapse microscopy and quantification of fluorescence

All live cell imaging experiments were performed at 37°C on a Nikon TI widefield microscope using epifluorescence illumination, a 40× 0.95 NA air objective and a Hamamatsu sCMOS camera. Cells were grown and imaged in 96-well glass bottom plates and 1 hr before imaging normal growth medium was replaced with DMEM:F-12 with HEPES and without phenol red, supplemented with 10% FCS and antibiotics. Microscopes were controlled by Micro-Manager software (*Edelstein et al., 2010, 2014*) and image analysis and quantification was performed using Micro-Manager and ImageJ. For image quantification, images were first corrected for unevenness in illumination using control images of a homogenously fluorescent slide. Images were then corrected for bleaching using the ImageJ plugin bleach correction (by J. Rietdorf). Fluorescent intensities were then measured in ImageJ and corrected for background. For single molecule FISH, Stellaris FISH probe sets for Emi1 were used (Biosearch technologies), consisting of 48 fluorescently labeled probes. Fixation and hybridization were performed as recommended by manufacturer.

## Cell synchronization and measurement of global translation

RPE-1 cells were synchronized by treatment with a CDK1 inhibitor, RO-3306, which blocks cells in G2 (see *Figure 1A*). Cells were either harvested at this point to obtain a G2 sample or the CDK1 inhibitor was removed to allow cells to enter mitosis. Upon removal of the inhibitor, ~50% of cells progressed into mitosis in a highly synchronous manner. At 45 min after RO release, mitotic cells were isolated using mechanical shake-off and either harvested to generate the mitotic sample or re-plated to allow mitotic exit and then harvested as a G1 sample 3 hr after re-plating. Global translation rates were measured using a 10 min incubation with S35-methionine. Cells were then washed and lysed and total protein was analyzed on a denaturing gel. Total radioactive S35 incorporation was quantified by measuring the intensity of the entire lane on the gel.

## Generation of a fluorescence-based translation reporter

We generated an inducible, fast-maturing green fluorescent protein (sfGFP) by fusing the inducible degron DHFR (*Iwamoto et al., 2010*) to sfGFP. An NLS was also added, concentrating the protein in the nucleus, which simplifies analysis and increases the signal-to-noise ratio. The DHFR-sfGFP-NLS protein was continuously degraded in the absence of the small molecule stabilizer, but rapidly accumulated upon stabilization by trimethoprim (TMP). Indeed, addition of TMP resulted in a rapid increase in GFP fluorescence, which was due to synthesis of new protein. Thus, the increase in GFP fluorescence is a good readout for the rate of translation of the reporter. However, different cells within a population usually contain different copy numbers of the reporter, and therefore showed different rates of GFP fluorescence increase. We therefore inserted a short viral P2A sequence after the GFP-NLS, which allows expression of a second independent protein from the same transcript with very high efficiency (>90%, [*Kim et al., 2011*]), due to the inability of the ribosome to form a peptide bond at the end of the P2A sequence. Downstream of the P2A sequence, we inserted a NLS-mCherry protein, which is insensitive to the degron, as the NLS-mCherry protein is physically separated from the DHFR-GFP-NLS protein. Levels of mCherry can thus be used to normalize for the total amount of reporter per cell (*Figure 3—figure supplement 1*).

## Cloning and plasmid sequences

Aurora A, Emi1, PP1γ and PP2aβ were PCR amplified from a cDNA library generated from RPE-1 cells. Aurora A was cloned upstream of GFP into the pHR lentiviral expression vector using MluI-NotI restriction sites (hereafter called pHR vector) with a truncated SV40 promoter to reduce expression levels. Emi1 was inserted downstream of mCherry in the pHR vector using BamHI-NotI. For the pHR-mCherry-P2A-PP1γ, mCherry was first PCR amplified and a P2A sequence was inserted into the 3′ primer with an extra RsrII restriction site. mCherry-P2A was then inserted in pHR using BstX1-NotI, after which PP1γ was inserted downstream of the P2A sequence with RsrII-NotI.

For pHR-mCherry-P2A-strepII-PP2aβ, mCherry was first PCR amplified with P2A-StrepII-tag sequence, as well as an extra 3′SpeI site in the 3′ primer. mCherry-P2A-strepII was then inserted into pHR, after which PP2aβ was inserted in this vector using SpeI-Not1. The fluorescence-based translation reporter was cloned using fusion PCR of three parts: 1. DHFR(Y100I), 2. sfGFP-NLS-P2A 3. NLS-mCherry. The product was cloned into the pHR vector using BstXI-NotI. All 5′ UTRs were inserted using BstXI-BsiWI and 3′UTRs were inserted using RsrII-NotI. Sequences of the entire reporter and all 5′ and 3′ UTRs used in this study can be found in the supplemental methods section. Primers to amplify the UTRs used in this study were based on the RNA-seq data to represent the most common UTR splice variant in RPE-1 cells.

## Acknowledgements

We thank Dr T Wandless for the DHFR(Y100I) plasmid. We would like to thank Nico Stuurman for help with microscopy and the Vale lab members for helpful discussions. MET was supported by fellowships from the Dutch Cancer Society (KWF) and EMBO (LTF 720–2011). NSG. was supported by a human frontiers science program postdoctoral fellowship. RDV. and JSW were supported by the Howard Hughes Medical Institute.

## Additional information

### Funding

| Funder | Grant reference | Author |
|---|---|---|
| Howard Hughes Medical Institute (HHMI) | | Jonathan S Weissman, Ronald D Vale |
| EMBO | LTF 720-2011 | Marvin E Tanenbaum |
| Human Frontier Science Program (HFSP) | | Noam Stern-Ginossar |
| KWF Kankerbestrijding | | Marvin E Tanenbaum |

The funders had no role in study design, data collection and interpretation, or the decision to submit the work for publication.

### Author contributions

MET, Conception and design, Acquisition of data, Analysis and interpretation of data, Drafting or revising the article; NS-G, JSW, RDV, Conception and design, Analysis and interpretation of data, Drafting or revising the article

## Additional files

### Supplementary files

• Supplementary file 1. Quality control of the deep sequencing data set. (**A**) The number of reads for the two biological replicates of the G2 FP sample was plotted on a log2 scale. Only genes with an average of at least 100 reads per replicate where selected for subsequent analysis. The box marked off by the two black lines indicates the genes that were excluded from the analysis, as they had less than 200 reads total. (**B**) After filtering the data (see 'Materials and methods') the ratio of the RPKM values was determined for all genes of each set of replicates. The standard deviation was determined for each set of replicates. The data sets of the two replicates were also fit and the $R^2$ for the fit was determined.

• Supplementary file 2. Genome-wide mRNA levels in G1, G2 and M phase cells.

• Supplementary file 3. Ribosome profiling dataset.

## Major dataset

The following dataset was generated:

| Author(s) | Year | Dataset title | Dataset ID and/or URL | Database, license, and accessibility information |
|---|---|---|---|---|
| Tanenbaum ME | 2015 | Regulation of protein translation during mitosis | http://www.ncbi.nlm.nih.gov/geo/query/acc.cgi?acc=GSE67902 | Publicly available at the NCBI Gene Expression Omnibus (Accession No: GSE67902). |

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
