## [Decision Letter]

[Editors’ note: this article was originally rejected after discussions between the reviewers, but the authors were invited to resubmit after an appeal against the decision.]

Thank you for choosing to send your work entitled “Regulation of protein translation during mitosis” for consideration at *eLife.* Your full submission has been evaluated by Vivek Malhotra (Senior Editor) and three peer reviewers, one of whom is a member of our Board of Reviewing Editor, and the decision was reached after discussions between the reviewers. Based on our discussions and the individual reviews below, we cannot accept the manuscript in its present form. If you can address the issues raised by the reviewers, we would gladly welcome a new manuscript for a formal review.

As you can see the reviewers agree that the data are interesting and the experiments were well performed and controlled. However, they raised substantive concerns about the interpretation of the data, especially in light of the activity of CDK1 as a kinase of 4EBP1 and eIF4G. In addition, you do not provide a good explanation why your data are at variance with the published literature. Nocodazole treatment cannot provide a simple explanation.

*Reviewer #1*:

This is an interesting manuscript by Tanenbaum et al. investigating the regulation of protein synthesis during mitosis. Utilizing RPE-1 cells and a CDK1 inhibitor to synchronize cells, the authors have assessed changes in mRNA levels (using RNA Seq) and translational efficiency (using Ribosome Profiling) during mitosis. They document a 35% reduction in 35S-methionine incorporation in mitosis and identify a set of transcripts whose translation appears to be more inhibited than others (as well as a few where translation is actually increased). The experiments appear well performed and the data, for the most part, support the drawn conclusions.

There are a few issues that could improve the quality of this manuscript.

1) Whereas a FACs analysis of the G2 sample is provided in Figure 1, what do representative analyses of the M and G1 samples look like?

2) Emi1 (aka FBXO5) (Gene ID: 26271) shows 3 transcripts with 3 different 5' UTRs. Which one(s) is/are expressed in RPE-1 cells? Which one was used to generate the reporter constructs? Which one is present during M phase? Do they all impart translational control? In fact, there seem to be 3 different proteins made via alternative exon 1 usage. Independent validation of the change in translation of the native transcript would significantly strengthen the author's conclusions here (and in Figure 4)

3) The Fluorescence-based translation reporter is interesting. Does trimethoprim affect mRNA levels? What is the efficiency of the P2A “Stop N Go” sequence (Figure 3)? For mCherry to be completely independent of the degron, it has to be close to 100%. Does one see GFP-mCherry fusions by Western blot?

4) Exactly which 5' UTRs were tested in Figure 4? If more than one 5' (or 3'UTR) is known for each specific transcript, then assaying only one out of a possible set of combinations needs to be justified (see my comment #2 above). For ARHGAP5 (Gene ID 394), there appear to be 3 different transcription start sites, which one was used in the Figure 4? Which one is relevant for RPE-1 cells? 5' UTR is fused to mCherry-Emi1 to render it under M phase control, but there are 3 different transcription start sites for this gene, which one was used?

Are the others also present in M phase? Do they impart translational control?

5) What is the efficiency of the P2A “Stop N Go” sequence (Figure 3)? For mCherry to be completely independent of the degron, it has to be close to 100%. If Westerns are performed, does one see GFP-mCherry fusions?

6) Does stabilization of mCherry-Emi1 produced from Emi1 or ARHGAP5 5'UTR prevent or delay the decrease in GFP-Aurora A seen at 120 mins?

*Reviewer #2*:

In this manuscript, Tanenbaum et al used Ribo-seq versus total RNA-seq analyses to explore cell cycle-dependent changes in protein synthesis rates at the whole genome level. Different pools of cells synchronized at different cell cycle phases were obtained by various declinations of experiments using a CDK1 inhibitor. G2-arrested cells were obtained by blocking cells with a CDK1 inhibitor for 18 hours, mitotic cells were obtained by a mitotic shake-off performed 45min following release from the CDK1 inhibitor block, and G1 samples were obtained by harvesting cells 3 hours after replating mitotic cells.

The authors explain that such technique of synchronization avoids the use of microtubule inhibitors which have been shown earlier to impinge upon translation rates and therefore may reflect an undesired side effect on protein synthesis rather than a cell cycle-dependent effect. Given the literature released after the original papers on this topic, I agree that the use of microtubule poisons is likely not a good way to explore mitosis specific translation.

The major concern I have with this paper is that CDK1 has been shown recently to directly phosphorylate two master regulators of protein synthesis, namely 4E-BP1 (Shuda PNAS, 2015) and eIF4GI (Dobrokov et al. Mol Cell Biol, 2014; 34:439-451), and that CDK1 phosphorylation of these two translation initiation components dramatically affects their translational activity. Therefore, the criticism against microtubule targeting drugs may also be true for CDK1 inhibitors. How can the authors be sure that the data obtained are not due to side effects on translation initiation instead of cell cycle-dependent events? This important comment applies to all the comparisons made with G2-arrested cells as they have been obtained by incubating cells with the CDK1 inhibitor for 18 hours. Unfortunately, the main message of the paper arises from comparisons between mitotic and G2-arrested cells.

Beside this important conceptual concern, the manuscript is very well written and data are clear and well presented. I however do not recommend publication in *eLife*. The manuscript must be revisited in light of CDK1 implication in protein synthesis through 4E-BP1 and eIF4GI phoshphorylation and activity.

*Reviewer #3*:

Protein synthesis rates fluctuate throughout the cell cycle and are dramatically reduced during G2/M transition. In this study, Tanenbaum et al. investigated regulation of mRNA translation during mitosis using ribosome profiling. They show that the mRNA levels in G2 and M phase cells are similar, but distinct from G1. In contrast, translation efficiency is similar in G2 and G1, but very different in M where the vast majority of mRNAs are translationally repressed. As the authors note, this conclusion is not novel as the decline in the rate of protein synthesis in eukaryotic cells during mitosis has been has been known for a very long time ([13] and the citations therein). The novelty here is in the methodology (ribosome profiling) and the conclusion for indiscriminate inhibition of translation of mRNAs during mitosis. The latter is in contrast to previously published data showing that ∼3% of mRNAs are associated with large polysomes and remain active during mitosis (41). Several other studies (mentioned in this article) have also demonstrated that internal initiation of translation is not inhibited during mitosis, in contrast to cap-dependent translation. Finally, IRES-mediated translation of poliovirus, mengovirus and human immunodeficiency type 1 virus is clearly not inhibited in mitotically-arrested (by nocodazole) cells, and these viruses replicate well in these cells, whilst viruses that rely on cap-dependent translation cannot replicate (Johnson and Holland, 1965, J Cell Biol. 27, 565-574; Lake et al., 1970, J Virol. 5, 262-263; [3], J Biol. Chem. 262, 11134-11139; Brasey et al., 2003. J Virol. 77, 3939-3949). It is not clear whether different cell lines, cell synchronization protocols or methods to evaluate translation efficiency (ribosome vs. polysome profiling) account for the observed differences. Given the significant discrepancies with published data, the authors should use other methodological approaches to support their conclusions. It is also puzzling why the authors were surprised by the results. Considering that only IRES-containing mRNAs, which constitute perhaps 3% of total mRNAs, are translationally active in mitosis, the observed results are expected rather than surprising. The Abstract and text should be changed to reflect this.

[Editors’ note: what now follows is the decision letter after the authors submitted for further consideration.]

Thank you for submitting your work entitled “Regulation of mRNA translation during mitosis” for peer review at *eLife.* Your submission has been favorably evaluated by Vivek Malhotra (Senior Editor), a Reviewing Editor, and two reviewers.

The reviewers have discussed their reviews with one another, and the Reviewing Editor has drafted this decision to help you prepare a revised submission.

In this second submission the authors have responded satisfactorily to some of the major concerns of the reviewers. However, some important comments (listed below) were not addressed:

1) The authors have presented new data indicating that the translational repression of Emi1 in mitosis “is present exclusively in the 3'UTR of Emi1.” Since 3' and 5' UTR elements can communication with each other, it is still important to know which mRNA isoform(s) the authors are dealing with. An effort should be made to map the 5' ends and get this information, as well as map the 3' end to ensure the UTR they are dealing with is relevant to the cell line (RPE-1) they are using. There is an underlying assumption that the Genbank Sequence is “full-length” AND the correct one for the cell line under study. However, this might not be true.

2) Original recommendation: “Independent validation of the change in translation of the native transcript would significant strengthen the author's conclusions here (and in Figure 4)”. The authors answer that ribosome profiling and the fluorescence reporter are sufficient to support the data. This is a bit of a circular argument. The fluorescence-based assay does not address what is happening with the native transcript. The initial discovery tool (ribosome profiling) does not validate the original finding.

3) Original comment: “What is the efficiency of the P2A “Stop N Go” sequence (Figure 3)? For mCherry to be completely independent of the degron, it has to be close to 100%. Does one see GFP-mCherry fusions by Western blot?” In lieu of providing a Western blot, the authors chose to cite Kim JH et al., 2011, PLoS One) (where P2A was tested in HEK293T, HT1080 and HeLa cells) and claim that it is 90% as previously reported. The assumption is that it is the same in RPE-1 cells and in their construct.

4) Original comment: “Exactly which 5' UTRs were tested in Figure 4? ” The authors responded: “Throughout this study we use the full length 5' and 3' UTRs”. 5' end heterogeneity exists due to alternative promoter and transcription start site usage. We still don't know if their reporters (Figure 3) contain the same 5' and 3' UTRs as present in RPE-1 cells on the native transcripts. Relying solely on GenBank sequences (as being “full length”) here just doesn't cut it. Given the critical role that sequences and/or structure in UTRs can have on mRNA output, these issues are important for the correct validation of the ribosome profiling results.

---

## [Author Response]

[Editors’ note: the author responses to the first round of peer review follow.]

We appreciate the constructive comments of the referees and have addressed them carefully in the revised manuscript. There were two major concerns with the manuscript: 1) whether the Cdk1 inhibitor used for cell synchronization in G2 phase might influence the conclusions of the study, given the known Cdk1 phosphorylation of 4EBP1 and eIF4G, and 2) why our data seems to be at variance with prior finding in the literature.

We regret that our prior manuscript was unclear on these two issues, but we feel that there are strong reasons to mitigate both concerns. A detailed discussion can be found in the response to the referees.

We feel that we have rigorously addressed the comments from the review process. We also feel that this study provides the first “big picture” view of translational regulation during mitosis and thus will be a well-regarded paper in *eLife*.

Reviewer #1:

*1) Whereas a FACs analysis of the G2 sample is provided in*
Figure 1*, what do representative analyses of the M and G1 samples look like?*

We have included new FACS analysis of sample purity for both M and G1 cells in Figure 1, which confirm that these samples too are very pure.

*2) Emi1 (aka FBXO5) (Gene ID: 26271) shows 3 transcripts with 3 different 5' UTRs. Which one(s) is/are expressed in RPE-1 cells? Which one was used to generate the reporter constructs? Which one is present during M phase? Do they all impart translational control? In fact, there seem to be 3 different proteins made via alternative exon 1 usage*.

The effects of differential isoform expression are interesting, and we have performed additional experiments to address this question. We have mapped the region in the UTRs of Emi1 that confers translational repression in mitosis and found that this activity is present exclusively in the 3’UTR of Emi1. Since the 3’UTR is identical in all Emi1 isoforms, this result suggests that all isoforms are most likely translationally controlled. We have now included this new data showing the involvement of the 3’UTR of Emi1 in Figure 3—figure supplement 1.

For clarity, we have also included the Emi1 UTR sequences (as well as the sequences of all other UTRs used in this study) in the Supplemental Methods section. The Emi1 3’UTR sequence that we used for our fluorescence translation reporter is identical to that of all known Emi1 isoforms (NM_001142522.1, NM_012177.3, XM_011535749.1) and importantly, matches the sequence of the 3’UTR of Emi1 as identified by our RNA-seq data (See new Figure 3—figure supplement 1). The 5’UTR used in our reporter assay (Figure 3) covers the complete 5’UTR of Emi1, and matches our RNA-seq data relatively well. Visual representation of these results are now included in Figure 3—figure supplement 1.

We also analysed isoform expression in G2 versus M, and found that the same Emi1 isoforms are expressed in both phases of the cell cycle (Figure 3—figure supplement 1).

*Independent validation of the change in translation of the native transcript would significantly strengthen the author's conclusions here (and in*
Figure 4*)*

We note that, in fact, we do have two independent lines of evidence that Emi1 is translationally repressed in mitosis. First, ribosome profiling revealed that the endogenous mRNA is translationally repressed. Second, we confirm these results using the fluorescence reporter and genetic elements of the Emi1 mRNA.

3) The Fluorescence-based translation reporter is interesting. Does trimethoprim affect mRNA levels?

Long term treatment of cells with trimethoprim does not affect mCherry levels, which indicates that the mRNA levels are not affected by trimethoprim treatment. This is confirmed by a previous study showing that trimethoprim has no detectable effects on mRNA levels ([19], Nat. Methods).

*What is the efficiency of the P2A* “*Stop N Go*” *sequence (*Figure 3*)? For mCherry to be completely independent of the degron, it has to be close to 100%. Does one see GFP-mCherry fusions by Western blot?*

The P2A sequence indeed confers separation of both protein with very high efficiency, generally around 90% or higher (Kim JH et al., 2011, Plos One). This is confirmed by the fact that mCherry levels are not detectably affected by TMP addition in our experiments (See Figure 3). A note has been added to the manuscript that refers to the high efficiency of the P2A sequence.

*4) Exactly which 5' UTRs were tested in*
Figure 4*? If more than one 5' (or 3'UTR) is known for each specific transcript, then assaying only one out of a possible set of combinations needs to be justified (see my comment #2 above). For ARHGAP5 (Gene ID 394), there appear to be 3 different transcription start sites, which one was used in the*
Figure 4*? Which one is relevant for RPE-1 cells?*

Throughout this study we use the full length 5’ and 3’ UTRs; the sequence of each is now provided in the Supplemental Methods section. Furthermore, we have now included a new figure (Figure 3—figure supplement 1), which shows the RNA-seq data across the Emi1 UTRs. Importantly, we have also included new data showing that the 3’UTR of Emi1 is responsible for translational repression (Figure 3—figure supplement 1). Since the 3’UTR is identical in all Emi1 isoforms (and matches the Emi1 3’UTR in our RNA-seq dataset (Figure 3—figure supplement 1), our new results show that all Emi1 isoforms are likely regulated in a similar fashion.

We have also included the sequence of the ARHGAP5 UTRs used in the experiments described in Figure 4. It is important to note that the goal of these experiments was to show that translational repression, rather than another function of Emi1’s UTR (like mRNA localization), was responsible for increased APC activation. The ARHGAP5 UTRs are therefore solely used as a tool to suppress Emi1 translation in mitosis (which they do very effectively, as shown in Figure 3), rather than to study ARHGAP5 translational regulation per se. In these experiments it is therefore not critical to know which ARHGAP5 isoforms are endogenously expressed in RPE1 cells. We have adapted the manuscript to indicate this more clearly.

Finally, as a general comment, we do acknowledge that it is difficult to exclude that specific isoforms may lack the genetic element that confers translational regulation. However, it is important to note that the level of translational regulation in our ribosome profiling is based on the sum of the reads over all mRNA isoforms. Thus, if the mRNA of a particular gene is translationally repressed on average by 5-fold in our dataset, it is possible that one isoform is repressed while another isoform is not regulated. However, this unregulated isoform cannot exceed 20% of the total mRNA for that gene, and therefore makes only a minor contribution to the overall synthesis of the encoded protein.

5' UTR is fused to mCherry-Emi1 to render it under M phase control, but there are 3 different transcription start sites for this gene, which one was used? Are the others also present in M phase? Do they impart translational control?

See our response to questions 2 and 4.

*5) What is the efficiency of the P2A* “*Stop N Go*” *sequence (*Figure 3*)? For mCherry to be completely independent of the degron, it has to be close to 100%. If Westerns are performed, does one see GFP-mCherry fusions?*

See our response to question 3.

6) Does stabilization of mCherry-Emi1 produced from Emi1 or ARHGAP5 5'UTR prevent or delay the decrease in GFP-Aurora A seen at 120 mins?

We have included a representative movie following GFP-Aurora A for several hours after mitosis, which reveal that mCherry-Emi1 under regulation of control UTRs prevents, rather than delays GFP-Aurora A degradation (Video 1). Aurora A is normally degraded (within 120 min) when Emi1 is under control of either Emi1 or ARHGAP5 UTRs.

Reviewer #2:

*[…] The major concern I have with this paper is that CDK1 has been shown recently to directly phosphorylate two master regulators of protein synthesis, namely 4E-BP1 (Shuda PNAS, 2015) and eIF4GI (Dobrokov et al. Mol Cell Biol, 2014; 34:439-451), and that CDK1 phosphorylation of these two translation initiation components dramatically affects their translational activity. Therefore, the criticism against microtubule targeting drugs may also be true for CDK1 inhibitors. How can the authors be sure that the data obtained are not due to side effects on translation initiation instead of cell cycle-dependent events? This important comment applies to all the comparisons made with G2-arrested cells as they have been obtained by incubating cells with the CDK1 inhibitor for 18 hours. Unfortunately, the main message of the paper arises from comparisons between mitotic and G2-arrested cells*.

We agree with the reviewer that this is a critical point for interpreting our study. However, we strongly believe that the use of this inhibitor does not impact the major conclusions of this manuscript, as described below. We have re-written the manuscript to clearly explain the implications of using this drug to synchronize cells.

1) CDK1 is indeed a critical kinase of several general translation factors, as described by the reviewer. However, throughout most of the G2 phase, CDK1 is completely inactive (Jackman et al., Nat. Cell. Biol., 2003; Gavet & Pines, Dev Cell, 2010); the initial activation of CDK1 occurs immediately prior (∼30 minutes) to mitosis (in prophase). Thus, our G2 sample, which is created by addition of a highly specific CDK1 inhibitor (53), actually mimics the normal G2 state very closely with respect to low CDK1 activity. Importantly, the notion that CDK1 is not active in phosphorylating general translation factors in G2, is fully supported by experimental data showing no significant phosphorylation of 4E-BP1 in G2 phase ([20], Curr. Biol).

When we release cells from G2 by removing the inhibitor with our synchronization protocol, then cells procede into M with high CDK1 activity (mitotic entry has an absolute dependence on CDK1 activation, so the fact that the cells enter mitosis in our synchronization protocol proves that CDK1 becomes fully activated (Lindqvist et al., JCB, 2009)). Phosphorylation of 4E-BP1 and eIF4GI by CDK1 then occurs during mitosis (Shuda et al., PNAS, 2015; Dobrikov et al., MBoC 2014) at a time when CDK1 activity is very high. These phosphorylation events could contribute to the effects on translation that we observed; however, these effects occur both in unperturbed M cells and the synchronized M cells in our study, which both have robust CDK1 activity. In summary, the use of a CDK1 inhibitor to arrest cells in G2 followed by its washout closely mimics the CDK1 levels in unperturbed cells in their G2, M and G1 states.

We apologize for the confusion due to the minimal description of our synchronization method and its implications in the initial version of the manuscript. We have now included detailed explanation of this critical point in this revised manuscript (in the Results and Discussion sections) and we have cited all the papers mentioned above.

2) A second important point is that the conclusions we draw based on our comparison between G2 and M cells are fully supported by our comparison of M and G1 cells, which both have no drug present. Not only are the genes that are repressed in M vs G2 largely the same as those repressed in M vs G1, the extent of repression is very similar too (new Figure 2). In both comparisons, it is clear that the vast majority of mRNAs that undergo specific regulation are translationally repressed in M, with very few mRNAs being translationally activated. We have performed new analysis that clearly demonstrates this important point (Figure 2) and we have adjusted the text accordingly.

3) We have performed independent follow-up experiments using the fluorescence-based translation reporter, which fully support the ribosome profiling results, at least on a subset of genes (Figure 3). Importantly, these follow-up experiments did not involve the CDK1 inhibitor, providing additional support that the observed effects are not due to the use of this inhibitor.

Reviewer #3:

*Protein synthesis rates fluctuate throughout the cell cycle and are dramatically reduced during G2/M transition. In this study, Tanenbaum et al. investigated regulation of mRNA translation during mitosis using ribosome profiling. They show that the mRNA levels in G2 and M phase cells are similar, but distinct from G1. In contrast, translation efficiency is similar in G2 and G1, but very different in M where the vast majority of mRNAs are translationally repressed. As the authors note, this conclusion is not novel as the decline in the rate of protein synthesis in eukaryotic cells during mitosis has been has been known for a very long time (*[13]
*and the citations therein). The novelty here is in the methodology (ribosome profiling) and the conclusion for indiscriminate inhibition of translation of mRNAs during mitosis*.

The global inhibition of translation of the large majority of mRNAs during mitosis has been known for a long time (13; 41). However, this “global” translational repression is conceptually distinct from the highly selective and *gene-specific* translational repression that we describe in this manuscript. We find (through S35met labelling) that the bulk of mRNAs show roughly a 30% decrease in translation in mitosis as compared to G2 or G1 (Figure 1), in overall agreement with earlier studies. This effect, which may be related to phosphorylation of general translation factors during mitosis by a mitotic kinase such as CDK1, is thus widespread and occurs indiscriminately and independent of mRNA sequence. However, we show for the first time that a small subset of mRNAs (1-2% of all mRNAs) deviates from the bulk, suggesting that these mRNAs undergo an additional, *gene-specific* regulation*.* This *gene-specific* regulation may occur through binding of RNA-binding proteins to the 5’ and 3’UTR; indeed the *gene-specific* nature of this regulation is confirmed by the fact that fusion of their UTR sequences to a reporter confers translational regulation to a report transcript. The unexpected result from our study is that this small group of *gene-specific* translationally regulated mRNAs (the 1-2% of all mRNAs) are mostly repressed in mitosis (indeed several fold more repressed than the 35% decrease that applies to virtually all mRNAs) while few show higher rates of translation. This *gene-specific* translation repression during mitosis was not previously known and thus is a major new finding of our study.

In summary, there are two distinct types of translational regulation that occur during mitosis:

“*Global*” translational repression that affects the bulk of mRNAs and likely occurs in a sequence-independent manner. This has been shown previously and simply confirmed here.

“*Gene-specific*” translational repression that affects ∼1-2% of mRNAs and acts on top of the global repression. This level of regulation has not been described previously and is a major new insight that arises from our study.

This distinction is a key point of our manuscript and we regret that we had not explained this point adequately. To correct for our error, we have used the definitions above throughout the text and have significantly re-written the manuscript in several places, including the Abstract. We feel that these changes have significantly improved the clarity of the manuscript.

*The latter is in contrast to previously published data showing that ∼3% of mRNAs are associated with large polysomes and remain active during mitosis (*[41]*). Several other studies (mentioned in this article) have also demonstrated that internal initiation of translation is not inhibited during mitosis, in contrast to cap-dependent translation*.

While a previous large-scale study ([41] which analyzed 1500 genes in contrast to our ∼8000 analyzed genes), as well as several “single gene” studies have indeed provided evidence that IRES-containing mRNAs can continue translation during mitosis, the notion that IRES-containing mRNAs are preferentially translated in mitosis has been contested by others recently ([46], PNAS; Stumpf et al., Mol Cell, 2013). In fact, in a very recent study, Shuda et al. (published as we submitted our manuscript) examined global cap-dependent versus cap-independent translation in mitosis in detail and found that mitotic translation was almost exclusively cap-dependent. Consistent with this, Stumpf et al. (Mol Cell, 2013) did not find translational upregulation in mitosis for the majority of the mRNAs that were previously thought to be translated in a cap-independent manner (including nucleophosmin, La and vimentin). Our genome-wide analysis is consistent with these recent papers, and suggest that IRES-mediated translation may not be the major driving force that shapes mitotic translation, as thought previously.

A possible explanation reconciling the above mentioned findings with earlier studies implicating IRES-mediated translation in mitosis, is that IRES-mediated translation during mitosis modestly enhances translation, but that this effect is small compared to the *gene-specific* translational *repression* that occurs during mitosis (which we identified in our study). Indeed, of the 3 genes [41] analyzed quantitatively (Nucleophosmin, La and vimentin), only one gene (Vimentin) showed more than 3-fold increase (∼4-fold) in translation in mitosis (note-3-fold is the cut-off used in our study). Thus, IRES-mediated translational stimulation in mitosis may occur, but is likely to be relatively minor compared to the substantial *gene-specific* translational repression that we identified in this study.

We realize that this is an important point of our study, and we made changes throughout our manuscript, including the Abstract, Introduction and Discussion to better explain our results in the context of previous studies implicating IRES-mediated translation in mitosis. We feel confident that this re-written text precisely places our manuscript in the correct context and explains apparent discrepancies with the literature.

*Finally, IRES-mediated translation of poliovirus, mengovirus and human immunodeficiency type 1 virus is clearly not inhibited in mitotically-arrested (by nocodazole) cells, and these viruses replicate well in these cells, whilst viruses that rely on cap-dependent translation cannot replicate (Johnson and Holland, 1965, J Cell Biol. 27, 565-574; Lake et al., 1970, J Virol. 5, 262-263;*
[3]*, J Biol. Chem. 262, 11134-11139; Brasey et al., 2003. J Virol. 77, 3939-3949)*.

Indeed, several studies have shown that viral IRESes promote robust translation initiation in mitosis. Here, we have focused on human mRNA translation though, and have not addressed viral translation in this study.

*It is not clear whether different cell lines, cell synchronization protocols or methods to evaluate translation efficiency (ribosome vs. polysome profiling) account for the observed differences. Given the significant discrepancies with published data, the authors should use other methodological approaches to support their conclusions*.

We agree that our study contradicts previous *thinking* on translational regulation during mitosis. However, as outlined in our response to point 2, our study does not necessarily contradict prior results per se, but rather reveals a new aspect of translational regulation that was not detected in earlier work that did not employ genome-wide techniques. We do not exclude the possibility that a small subset of mRNAs are translated more efficiently in mitosis due to the presence of IRESes, and have added this statement to the revised manuscript. However, our genome-wide analysis (the first of its kind to accurately quantify both translational activation and repression in mitosis) shows that the dominant effect of *gene-specific* translational regulation involves translational repression, rather than activation. Thus, previous cases of IRES-dependent translational activation of a few specific mRNAs during mitosis appear to be exceptions rather than the rule for most mRNAs. Furthermore, it is important to note that we have provided a completely independent validation of our study through the use of a single cell, fluorescence-based translation reporter (Figure 3), as suggested by the reviewer.

Overall, we rewrote our manuscript to explain our new results in the context of prior studies (Abstract and Discussion).

*It is also puzzling why the authors were surprised by the results. Considering that only IRES-containing mRNAs, which constitute perhaps 3% of total mRNAs, are translationally active in mitosis, the observed results are expected rather than surprising. The Abstract and text should be changed to reflect this*.

This comment stems perhaps from confusion over the manner in which we have presented our results (see our comments to points 1, 2 and 4). We have now re-written several parts of the manuscript (including the Abstract) to clarify this issue.

[Editors’ note: the author responses to the re-review follow.]

*1) The authors have presented new data indicating that the translational repression of Emi1 in mitosis* “*is present exclusively in the 3'UTR of Emi1.*” *Since 3' and 5' UTR elements can communication with each other, it is still important to know which mRNA isoform(s) the authors are dealing with. An effort should be made to map the 5' ends and get this information, as well as map the 3' end to ensure the UTR they are dealing with is relevant to the cell line (RPE-1) they are using. There is an underlying assumption that the Genbank Sequence is* “*full-length*” *AND the correct one for the cell line under study. However, this might not be true*.

We agree that the Genbank sequence may not reflect the sequence of transcript isoforms expressed in RPE1 cells. However, we did not base the UTR sequence used in Figure 3 and Figure 4 on the Genbank sequence. Rather, we carefully examined our RNA-seq data of RPE1 cells and cloned the Emi1 UTR sequences that most closely resembled the actual Emi1 isoform UTR sequence as expressed in RPE1 cells. The Emi1 UTR sequence expressed in RPE1 cells, the Genbank sequences, and the UTR sequences used in the reporter were all presented in Figure 3—figure supplement 1 (now Figure 3—figure supplement 1) of our previous submission to highlight this point.

Nonetheless, to completely exclude any isoform specific effects, we have now tested a set of new reporters covering two additional possible Emi1 5’UTRs and found that all 5’UTRs confer mitosis-specific translational repression when combined with the 3’UTR of Emi1. Thus, these new results agree with our prior conclusion that the translational regulation is conferred by the 3’UTR sequence of Emi1 and that different isoforms at the 5’UTR do not alter this effect to any significant extent. Together, these new experiments, combined with our previous revision, provide very strong support for the role of Emi1’s UTRs in translational repression in mitosis.

*2) Original recommendation:* “*Independent validation of the change in translation of the native transcript would significant strengthen the author's conclusions here (and in*
Figure 4*)*”*. The authors answer that ribosome profiling and the fluorescence reporter are sufficient to support the data. This is a bit of a circular argument. The fluorescence-based assay does not address what is happening with the native transcript. The initial discovery tool (ribosome profiling) does not validate the original finding*.

We feel that our evidence for Emi1 translational repression is strong. The discovery tool (ribosomal profiling) is the most sensitive and accurate tool for measuring ribosome occupancy. Neither state-of-the-art ribosome profiling nor related methods of ribosome occupancy (e.g. polysome analysis by sucrose gradient) provide a direct measurement of translation, which is a potential caveat. For these reasons, we used a direct translation readout of a reporter construct containing the Em1 sequence. While the reporter is not precisely the same mRNA as the native construct, it yielded the same conclusion (a very large translational repression) and thus was in good agreement with the ribosome profiling result. Thus, we feel that these are two independent and reasonable arguments for translational regulation, and thus we would not necessarily apply the word “circular” to the logic. One argument measures native transcript + an indirect measure translation (ribosome occupancy) and the other measures a reporter construct + a more direct measure of translation (fluorescence readout from the reporter). Both of these experimental strategies yielded very similar quantitative readouts. Given that ∼1% of genes show translational repression in mitosis, this agreement of these results is unlikely to be spurious. We observed similar good agreement for other mRNAs between ribosomal profiling and the other translational reporter results and also tested further isoforms for translational repression in mitosis (Figure 3—figure supplement 1).

We have now laid out these arguments very clearly and directly in the Discussion. In fact, we have rewritten the Discussion so that it opens with a new section that presents the potential caveats raised through the review process: 1) methodologies used this paper, and 2) the use of the CDK1 inhibitor (addressed in the earlier revision). These issues are now presented front-and-centre along with our reasons why we feel that we have a compelling case for *gene-specific* translational control in mitosis. Thus, the reader is explicitly made aware of these points before proceeding to higher level discussions of the significance of the results.

*3) Original comment:* “*What is the efficiency of the P2A* “*Stop N Go*” *sequence (*Figure 3*)? For mCherry to be completely independent of the degron, it has to be close to 100%. Does one see GFP-mCherry fusions by Western blot?*” *In lieu of providing a Western blot, the authors chose to cite Kim JH et al., 2011, PLoS One) (where P2A was tested in HEK293T, HT1080 and HeLa cells) and claim that it is 90% as previously reported. The assumption is that it is the same in RPE-1 cells and in their construct*.

We have performed a western blot on cells expressing our reporter and find that the efficiency of the P2A ribosome skipping sequence is around ∼85-90% (Figure 3—figure supplement 1), in close agreement with previous reports.

*4) Original comment:* “*Exactly which 5' UTRs were tested in*
Figure 4*?* ” *The authors responded:* “*Throughout this study we use the full length 5' and 3' UTRs*”*. 5' end heterogeneity exists due to alternative promoter and transcription start site usage. We still don't know if their reporters (*Figure 3*) contain the same 5' and 3' UTRs as present in RPE-1 cells on the native transcripts. Relying solely on GenBank sequences (as being* “*full length*”*) here just doesn't cut it. Given the critical role that sequences and/or structure in UTRs can have on mRNA output, these issues are important for the correct validation of the ribosome profiling results*.

We apologize for not pointing this out more clearly in our manuscript and previous response letter, but again (see our response to point 1), we did not rely on GenBank sequences, but designed the translation reporters based on the isoforms that are actually expressed in RPE1 cells (as determined by our RNA-seq data), exactly as the reviewers suggest. We have provided new figures (Figure 3—figure supplement 1) that present the RNA-seq. data together with a schematic overview of the UTR isoforms used in our reporters. This new analysis shows that the reporters used here closely mimic the endogenous transcripts.